# Blood Transcript Biomarkers Selected by Machine Learning Algorithm Classify Neurodegenerative Diseases including Alzheimer’s Disease

**DOI:** 10.3390/biom12111592

**Published:** 2022-10-29

**Authors:** Carol J. Huseby, Elaine Delvaux, Danielle L. Brokaw, Paul D. Coleman

**Affiliations:** 1ASU-Banner Neurodegenerative Disease Research Center, Arizona State University, Tempe, AZ 85281, USA; 2Perelman School of Medicine, University of Pennsylvania, Philadelphia, PA 19104, USA

**Keywords:** machine learning, Alzheimer’s disease, Huntington’s disease, Parkinson’s disease, amyotrophic lateral sclerosis or Lou Gehrig’s disease, frontotemporal dementia, Friedreich’s ataxia, linear discriminant analysis, whole blood RNA, random forest classification

## Abstract

The clinical diagnosis of neurodegenerative diseases is notoriously inaccurate and current methods are often expensive, time-consuming, or invasive. Simple inexpensive and noninvasive methods of diagnosis could provide valuable support for clinicians when combined with cognitive assessment scores. Biological processes leading to neuropathology progress silently for years and are reflected in both the central nervous system and vascular peripheral system. A blood-based screen to distinguish and classify neurodegenerative diseases is especially interesting having low cost, minimal invasiveness, and accessibility to almost any world clinic. In this study, we set out to discover a small set of blood transcripts that can be used to distinguish healthy individuals from those with Alzheimer’s disease, Parkinson’s disease, Huntington’s disease, amyotrophic lateral sclerosis, Friedreich’s ataxia, or frontotemporal dementia. Using existing public datasets, we developed a machine learning algorithm for application on transcripts present in blood and discovered small sets of transcripts that distinguish a number of neurodegenerative diseases with high sensitivity and specificity. We validated the usefulness of blood RNA transcriptomics for the classification of neurodegenerative diseases. Information about features selected for the classification can direct the development of possible treatment strategies.

## 1. Introduction

Neurodegenerative diseases such as Alzheimer’s disease (AD) are complex and require expensive diagnostic procedures. In addition to the expense, a recent study found AD has a misdiagnosis rate of about 1 in 4 even at established dementia centers [1]. Because AD symptoms often overlap with common disorders for which many are treatable or reversible, an accurate diagnosis at the earliest time is crucial [2]. The use of multiple research biomarkers coupled with clinical evaluations has improved diagnostic accuracy for neurodegenerative diseases, such as magnetic resonance imaging (MRI) and positron emission tomography (PET), including 18F-fluorodeoxyglucose (FDG) assessment of glucose metabolism, measurement of Aβ plaque burden in the brain using PIB, and tau neurofibrillary tangle burden with flortaucipir 18F. Additional biomarker tests include cerebral spinal fluid (CSF) assessment of Aβ clearance, total tau levels, and phosphorylated tau [3,4]. These types of measurements are expensive and invasive with substantial health risks such as nerve damage, infection, and radioactive tracers. In some cases these tests are not covered by standard health care [5].

Neurodegenerative diseases are associated with changes in specific molecular pathways and involve metabolomic and biological processes active in both the central nervous system (CNS) and vascular peripheral system [6]. The study of blood-based changes in mRNA gene expression presents a good strategy for differentiating patients of any neurodegenerative disease regardless of the proteins or their posttranslational modifications occurring in disease. Using RNA expression as blood biomarkers can also provide information for a pathophysiological relationship with disease and gives us the advantage of using the vast knowledge of gene expression to not only predict disease but also analyze the pathways affected. With expression platforms of more than 30,000 probes covering 10,000 genes, a blood test can be constructed to not only rule out AD but also indicate which dementia may be at work.

Blood biomarkers are especially interesting having low cost, minimal invasiveness, and accessibility to almost any world clinic. The incorporation of blood screening tests in clinics around the world, which, ideally, could also classify neurodegenerative diseases, could provide valuable support for clinicians when combined with cognitive assessment scores, thereby reducing health care costs and ruling out or streamlining additional tests such as imaging. Another ideal utility of an inexpensive blood panel is the ability to monitor disease progression, for example, marking the advancement beyond mild cognitive impairment (MCI). Identifying blood-based biomarkers for AD is highly attractive and will prove essential for screening, diagnosis, differentiating, and monitoring or predicting disease progression and therapeutic outcome. Screening tests are already available for various disorders, such as gestational diabetes, cancers, and depression to name a few, but the majority are limited to simply ruling out disease having low positive predictive power [7,8,9,10]. A simple, cost-effective blood screening test for neurodegenerative disease with high negative predictive value may not be a positive predictor of disease but can reduce health care costs by decreasing the amount of patient evaluation time and ruling-out whether a patient needs additional dementia tests such as expensive imaging procedures. Such a test might also reduce the cost of clinical trials by refining the selection of patients to the trial [11,12,13,14].

Our previous blood transcriptome work was able to distinguish AD, PD, and control brains and was found to be reproducible across multiple replicates and platforms [15]. The transcript selection was hypothesis driven and evaluated on custom cDNA arrays and qPCR. We found significant success using multivariate analysis on our previous transcript predictors belonging to cell stress, inflammation, and epigenetics functional groups in peripheral leukocytes as a minimally invasive diagnostic tool. We continued our analysis of these early transcript predictor picks and utilized machine learning techniques to discover groups of transcripts that may classify disease but also guide our discovery of affected pathways. Machine learning techniques based on the classification of transcriptional expression data between multiple diseases and/or non-disease have been successfully applied to differentiate cancer types and subtypes such as that encountered in the large group of small-blue-round-cell tumor cancers [16,17,18]. Information about features selected for the classification may also guide insight into possible treatment strategies. Publicly available whole blood expression datasets of six neurodegenerative diseases were selected from the Gene Expression Omnibus database for analysis including AD, idiopathic Parkinson’s Disease (PD), amyotrophic lateral sclerosis or Lou Gehrig’s disease (ALS), Friedreich’s ataxia (FRDA), behavioral variant frontotemporal dementia (FTD), and Huntington’s Disease (HD). Our choices were limited to large whole blood transcript datasets publicly available at the time of this study. The selected transcript predictors were then subjected to a test of classification by linear discriminant analysis (LDA) for each disease.

Here we aim to validate our previous transcript picks using new datasets and we find that the complete set of transcripts performs equally well or better at discriminating between clinical AD and age/sex-matched nondemented individuals, providing again proof of concept of the practical use in a blood screening test. We expanded our transcript discovery using a random forest algorithm to derive supervised predictors of different neurodegenerative diseases with the aim of generating transcriptional clusters able to make clinical group discriminations that are unique to each disease. RNA expression analysis gives us the advantage of using the vast knowledge of gene expression to classify disease.

## 2. Materials and Methods

### 2.1. Data Aquisition

The gene expression data used here were sourced from the publicly available repository Gene Expression Omnibus (GEO) (https://www.ncbi.nlm.nih.gov/geo/) in June, September, November, and December 2020. The datasets were chosen using the following criteria: (1) RNA extracted from whole blood collected in Paxgene tubes; (2) dataset must contain at least one hundred human subjects; and (3) data were generated on the GPL15988nuID (Illumina HumanHT-12 V4.0 expression beadchip), GPL10558 (Illumina HumanHT-12 V4.0 expression beadchip [19,20,21]), GPL6947 (Illumina HumanHT-12 V3.0 expression beadchip [20,22,23]), or GPL570 ((HG-U133_Plus_2) Affymetrix Human Genome U133 Plus 2.0 [24]) microarray expression platforms. These platforms contain more than 20,000 gene probes and provide overlapping transcripts to ensure comparison across models generated. Meta-analysis was performed on ten cohorts (GEO datasets GSE63060 (AD1) [20,25], GSE63061 (AD2) [20], GSE140829 (AD3), GSE57475 (PD1), GSE99039 (PD2 and HD) [24], GSE112676 (ALS1) [19,23], GSE112680 (ALS2) [19,23], GSE102008 (FRDA) [21,26], and GSE140830 (FTD) [27]) with a total of 3490 samples [19,20,21,22,23,24]. Cohort details are tabulated in Appendix A.

### 2.2. Dataset Normalization

Blood RNA expression datasets were downloaded and processed in normalized form. Within each dataset we used each sample’s respective GUSB (accession NM_000181.2) expression level to ensure normalization of samples was identical to that used for normalization in the previous blood work. GUSB is commonly used in laboratory experiments to normalize data and is found to be most consistently unchanged in neurodegenerative disease [15,28,29]. To standardize the transcript selection across all datasets, transcript matching and sorting were conducted by specific platform probe identifier when possible to ensure that each random forest (RF) analysis was conducted on identical transcript lists. We anticipated platform-specific effects would be present as well as batch effects for each cohort. Both LDA and RF analysis were performed on data from each platform separately. Mapping between Illumina and Affymetrix platforms on accession number is problematic due to multiple accession numbers for which probes only exist on one platform [30]. For this reason, we mapped between platforms from different manufacturers at the gene level. Without an exact sequence match, all probes identified as one EntrezID were then averaged.

### 2.3. Linear Discriminant Analysis (LDA)

The normalized data values for the predicting sets of transcripts were analyzed by multivariate discriminant analysis GB-STAT v.10 (Dynamic Microsystems, Inc., Silver Springs, MD, USA.). Linear discriminant analysis is a generalization of Fisher’s linear discriminant and maximizes the separability among the two categories. A linear discriminant function (weighted sums of the transcripts) was computed for maximum separability of the two groups where the coefficients or discriminant scores are used to classify as disease or controls.

### 2.4. Random Forest Classification (RF)

A random forest classifier is a machine learning algorithm that builds an ensemble of many classifier trees, where the prediction for a test sample is made by majority vote tallied on a combination of all trees [31,32]. We developed a new classification algorithm using a modification on the Matlab (Mathworks, R2020b) implementation of RF, a machine learning algorithm from the work of Leo Breiman and others [31,33,34]. Each individual tree is built from a training set of *n* random samples (with replacement) from *N* available samples referred to as bootstrap sampling. Predictions gathered from a collection of trees using bootstrap sampling is called bagging, thus the Matlab function *treebagger.* The bootstrap sampling selection procedure results in a random exclusion of approximately a third of the samples from the training set. These samples, referred to as ‘out-of-bag’ samples, are used later to calculate an estimate of the generalization error. All of the remaining two-thirds of the training samples called ‘in-bag’ samples are randomly selected with replacement to grow each new tree with at least one observation per tree leaf. The maximum depth of a tree is set to 20. Blood RNA transcripts’ or features’ expression values for a small subset (f) selected from all transcript features (F), $f=\sim \sqrt{F}$, are randomly selected (without replacement) to partition each binary node in the tree using the weighted Gini impurity index $\left(1-\overset{n}{\underset{i=1}{\sum }}p_{i}^{2}\;\right),$ which is an estimate of the least impurity for likelihood of misclassification [31,33]. This index is used to rank the relative importance of features for sample classification [34]. Many trees are built in this way before the remaining 1/3 of the out-of-bag training samples are used to select the best decision tree. In our model, the top-ranked transcripts from the training session were applied to the independent test sample set to tally the four possible accuracy outcomes in 2X2 confusion matrix format. One hundred classifying sessions were made in which the number of trees in the forest or estimators changed in a four-step process (Figure 1). A new random seed was generated before each session and the top session with greatest sensitivity was selected. Step (1) build 1000 trees, Step (2) select top 500 classifying features, Step (3) build 5000 trees using top 500 ranked classifiers, and Step (4) select and save top 20 classifiers for each 100 sessions including confusion matrix generated from independent test set validation. Repeat steps 1–4 each time using a new random seed to ensure complete feature selection randomization.

Random forest classifications for individual datasets were made by random 80/20 separation of a training and test set where 20% of the samples from each class were removed prior to analysis and used for the validation test step after random forest classification and selection of transcript predictors. For the selection of AD and ALS predictors, we were able to use whole datasets (i.e., GSE63060 (AD1) with GSE63061 (AD2) and GSE112676 (ALS1) with GSE112680 (ALS2) whole blood expression data), one set was used for training, while the other set for the independent test data and vice versa.

## 3. Results

### 3.1. Literature-Based Transcript Selection for Stress Response Classifies Disease

Our overall objective was to discover a small set of transcript predictors in whole blood that are reliable low-cost identifiers of AD. We also sought to compose from our meta-analysis, a set of classifying biomarkers that distinguish between neurodegenerative diseases, even those that may have similar clinical symptoms. We extended our evaluation of a group of prediction transcripts or features compiled previously where the feature selection was guided by evidence-based empirical data and the AD literature available at that time [15,35]. Twenty-two transcripts previously selected [15] (Table 1), representing genes from three different functionally relevant mechanisms in AD (inflammation, epigenetics, and stress), were subjected to linear discriminant analysis (LDA) on expression values collected from whole blood in eight new cohorts across six neurodegenerative diseases including AD, PD, ALS, FRDA, FTD, and HD. On the basis of prediction power as a percentage of the area under the curve (AUC) constructed from LDA discriminative scores assigned to each sample, our previous selection of seven inflammation transcripts made a poor classification (AUC<=70%) for any of the disease cohorts tested here [36,37] (Figure 2a). The six epigenetics transcripts were also poor classifiers of disease from controls (Figure 2b); however, the nine transcripts associated with stress processes did generate acceptable predictive percentages (AUC>70%) for both ALS1/2- (AUC 77%, AUC 75%) and HD- (AUC 75%) affected cohorts (Figure 2c). Analyses using twenty-two transcripts including all three functional groups increased the predictive power of disease and successfully classified disease from controls for three of the neurodegenerative diseases AD1 (AUC 80%), AD2 (AUC 72%), ALS1 (AUC 79%), ALS2 (AUC 80%), and HD (AUC 75%) but not for PD1 (AUC 67%), PD2 (AUC 67%), FTD (AUC 70%), or the FRDA group (AUC 68%) (Figure 2d).

Inspection of the significance and separation between the classes was measured as Wilks’ Λ, a ratio of the class variations (Λ=|E|/|H+E|) where *E* is the determinant of a matrix of variations within classes and *H* the determinant of a matrix of variations between the two classes of disease samples and controls. A number close to zero with a significant *p*-value suggests the LDA made an excellent separation of the classes distinguishing disease samples and controls. We found that for the groups of transcripts that were able to classify disease from controls more than 70% of the time (7 stress transcripts and all 22 transcripts) that the Wilks’ Λ was in general less than 0.8 with F-ratio significant *p* < 0.05 in agreement with successful AUC values reported. An overview of LDA analysis statistics across the six neurodegenerative diseases tested here for the three functional groups of previously selected transcripts is tabulated in (Appendix A).

### 3.2. Machine Learning Selection of Transcripts in Blood Classifies AD

We further sought to find new sets of blood transcript disease classifiers using a novel supervised machine learning algorithm. The random forest classifier randomly selects two-thirds of the samples with replacement in the training set and orders them by the expression of each *f* randomly selected transcript feature without replacement for each node in the binary classification tree. The Gini index is used to select transcripts at each node with the least impurity to build trees and thus ranks the transcripts by importance. After building many trees in this way, the remaining one-third of the training samples are used to select the best decision tree. Finally, a confusion matrix is generated using a separate independent test sample set. One AD blood expression dataset is used for the supervised training set, while the other AD dataset samples are used to validate and measure the predictive power of the transcripts chosen in the learning session. To ensure the best possible scenario for feature selection overlap between the two expression datasets, we first matched and kept only feature probes that appeared in both AD whole blood sets [20,38]. About 2000 unidentified gene probes were removed from both datasets, resulting in 23,240 matched transcript probes for 329 samples in AD1 and 388 samples in AD2, both containing a mix of AD, MCI, and HC cases. Our analysis did not include the MCI cases and six samples were excluded from further analysis in AD2 due to ambiguous clinical classification [38].

The top twenty predicting transcripts, matching probe IDs, and classification scores were saved for one hundred rounds of the four-step random forest classification scheme described in ‘Materials and Methods’ (Figure 1). Selecting from the one hundred models generated, the top twenty features with the best sensitivity score were then used as predictors for further analysis. Expression data in AD1 were used as the training set and AD2 the independent test set, resulting in a set of transcript disease predictors (Table 2, AD1) with a sensitivity performance of 78%, while training on data in AD2 and testing on AD1 generated a second set of transcript disease predictors (Table 2, AD2) with disease prediction sensitivity of 81%. Six transcripts (*MRPL51, NDUFA1, NDUFS5, two Illumina probes for RPL36AL, and LOC646200 (similar to 60S ribosomal protein L22, heparin-binding protein HBp15, NCBI record removed)* were selected in both training sessions of random forest analysis on the AD blood datasets. These six transcripts selected consistently for AD classification represent the electron transport chain and ribosomal complexes.

Each of the two groups of twenty RF-selected transcripts, which discriminate between AD and controls, were further analyzed for classification power in other diseases using linear discriminant analysis (LDA). Whole blood gene expression datasets from ten cohorts spanning six neurodegenerative diseases including AD, PD, ALS, FRDA, FTD, and HD were analyzed. We first measured a baseline prediction probability for the LDA by the random number generator selection of twenty transcripts in both AD1 and AD2. We found that randomly selected transcripts can classify AD from healthy controls (HC) (AUC ≤ 71%), further justifying that good classification should be well above AUC > 71% (Appendix A). The group of features selected when training on AD1 and subsequently validated in AD2 successfully classified AD, ALS, and HD from controls with excellent accuracy [37], making discriminatory probabilities (AUC > 80%) for the AD1 test set (AUC 82%), ALS1 (AUC 83%), ALS2 (AUC 87%), and HD (AUC 83%). The classification for a third AD blood dataset was found to be acceptable (GSE140829AD, AUC 71% data not shown), while the classification within PD, FRDA, and FTD from controls was poor with probabilities for PD1 (AUC 67%), PD2 (AUC 67%), FRDA (AUC 63%), and FTD (AUC 70%) showing this set of transcript predictors has discriminating utility for classifying AD (Figure 3a). The group of similar features selected when training on AD2 and subsequently validated in AD1 again classified AD and HD with excellent discriminating probability for AD1 (AUC 87%) and HD (AUC 80%), but in the case of ALS, this random forest transcript set picks (ALS1 and ALS2, AUC ~70%) did not repeat the excellent performance shown by the former transcripts selected by training on AD1. This is probably due to the reduced number of AD transcript predictors (Figure 3b) present in the two ALS sets. Sixteen of the former set of transcripts were present in the ALS blood datasets but only nine of the second set of twenty transcripts were present in the ALS blood datasets. Interestingly, in the case of HD, only eleven of this second set of AD-predicting transcripts were present, but these eleven were able to discriminate HD from controls with excellent performance. The second set of twenty features from training on AD2 were also able to classify FTD from controls with acceptable performance (AUC 74%). Classification performance testing on the third AD blood expression dataset was acceptable (GSE140829AD, AUC 72%, data not shown), while classification within PD and FRDA was less than acceptable for PD1 (AUC 64%), PD2 (AUC 68%), and FRDA (AUC 64%) (Figure 3b).

Inspection of the significance and separation between classes (disease and control) measured by the Wilks’ Λ was in good agreement with the AUC probabilities. Again, a Wilks’ Λ less than 0.8 represented good class separation combined with an F-ratio *p* < 0.05 and resulted in at least 75% of the case samples being classified correctly. The LDA statistics for random forest AD transcript predictors are documented in Appendix A.

### 3.3. Machine Learning Selection of Blood Transcript Classifiers for other Neurodegenerative Diseases

We further sought to select sets of RF classifiers for each of the other neurodegenerative diseases. We acquired two datasets each of ALS and PD; however, the two PD datasets were collected and generated on different manufacturer platforms by different research groups and we found it difficult to pair up and match transcripts between the cohorts due to different lengths and binding sites of oligomer probes and subsequently chose to treat the two PD sets individually. The random forest classifier method was applied to the blood RNA expression datasets to distinguish between disease and healthy controls. In the case of ALS, the expression datasets were first matched and sorted against each other to ensure each contained the same set of features.

The random forest classification algorithm generated one hundred models in a four-step process as described in ‘Materials and Methods’ (Figure 1), and the top twenty transcript predictor names, probe IDs, and scores were saved for each model. The best prediction model measured within the RF algorithm for sensitivity and specificity was selected out of the one hundred models generated for further analysis. For ALS, as previously stated, the training was performed on one dataset while using the other dataset as the test samples. In all other diseases, a test set of samples was first separated from the whole for later use in the validation step of the random forest. The test sample set was selected by randomly removing 20% of each of the control and disease samples. The remaining 80% of the data was used in the training steps of the RF algorithm.

Training on ALS1 and test data ALS2 resulted in a set of 20 transcripts with sensitivity 74%, while training on ALS2 and testing on ALS1 generated a set of 20 transcripts (Table 2, ALS1 and ALS2) with disease prediction performance in the test set of 91% sensitivity. The PD data cohorts were treated individually by first separating out 20% of the test samples and the resultant RF-selected 20 predictive transcripts (Table 2, PD1 and PD2) each performed for PD1 at 89% sensitivity, and for PD2, the top set of predictors only gave the best performance at an RF sensitivity of 60%; however, when subjected to downstream validation using LDA, the 20 transcripts were able to classify PD from controls with 83% accuracy (Appendix A). There was only one cohort each of HD, FTD, and FRDA expression data and all were first divided into training and test sets. For HD, which is a cohort within the PD2 dataset (GSE99039), the 20 classifying transcripts (Table 2, HD) selected on the training dataset and validated in the test set measured an excellent performance of the best model at 83% sensitivity. The RF selection on FTD and FRDA generated twenty transcript classifiers each with 71% sensitivity and 79% sensitivity, respectively (Table 2, bvFTD and FRDA).

For each disease cohort dataset, the respective group of twenty transcripts selected by RF to discriminate between disease and controls were further validated using LDA on the test samples set aside for the validation step. The results for AD groups (AD1 and AD2) are plotted together in Figure 3a,b (black lines) with excellent discrimination for both RF transcript sets, AD1 picks (AUC 82%) and AD2 picks (AUC 87%). The group of features selected for each of the ALS cohorts classified disease and control samples with excellent accuracy (ALS1, AUC 91%; ALS2, AUC 90%) (Appendix A). Similarly, features selected on training sets with PD cohort datasets and validated using the isolated test sets generated exceptional accurate classification (PD1, AUC 97%; PD2, AUC 83%) (Appendix A) using multivariate LDA analysis. In the case of HD, training generated 20 transcripts; however, the number of samples in the test set was limited by the size of the disease cohort, and only 10 of the twenty transcripts picked could be used for LDA. A selection of the top 10 ranked features by the RF were used in the analysis on the test set, which, regardless of the reduced number of transcripts, had excellent classification (AUC 89%) (Appendix A). The LDA analysis for both FTD and FRDA also provided good classification (FTD, AUC 75% and FRDA, AUC 80%) (Appendix A).

An inspection of separation statistics (Appendix A) suggests that although the discriminant analysis of GSE57475PD (Wilks’ Λ = 0.13), GSE99039HD (Wilks’ Λ = 0.49), GSE140830FTD (Wilks’ Λ = 0.70), and GSE102008FRDA (Wilks’ Λ = 0.55) implies good classification separation by LDA for small Wilks’ Λ, the F-ratio *p* > 0.05 for each indicates LDA separation may be due to chance only. Additional testing of the selected transcripts will be required for validation.

## 4. Discussion

An inexpensive and minimally invasive blood test is ideal for earlier diagnosis of people at risk of neurodegeneration and for gaining early intervention for testing and treatment strategies. Considerable time and resources are required to evaluate patients for neurodegenerative diseases or to screen research participants using current expensive and invasive methods such as PET scans and spinal taps. An accurate, inexpensive, noninvasive blood test can rapidly screen large and diverse groups from clinics around the world for diagnosis or recruit volunteers for studies [39]. Promising blood tests emerging for AD include blood tests measuring the level of phosphorylated tau protein [4] or Aβ species [40] indicating a neurodegenerative disorder in brain. The advantage of using blood RNA expression is it encompasses the entire transcriptome rather than a single or limited number of protein species specific for disease that may not appear until later after irreversible damage has already occurred. Using the entire transcriptome also has utility for a broader application to multiple diseases. By examining the complex RNA landscape in blood, we capture heterogeneities that can rule out disease or identify early markers indicating the presence of processes leading to neurodegenerative disease even before extensive neuron damage and clinical symptoms begin to develop. Our overall objective is to work toward discovering a small set of transcript predictors in whole blood for reliable, low-cost identification of dysfunctional processes leading to AD. We used a novel RF algorithm to select transcripts that were able to differentiate disease from controls but also found that these transcripts could classify other neurodegenerative diseases and controls revealing similarities and differences between diseases.

Previously, we described success for LDA to distinguish AD from healthy controls and PD using the 22 empirically-selected transcripts [15]. We extended the validation analysis to include new blood expression datasets and found these transcripts satisfactorily predicted disease from healthy controls in the cases of AD, ALS, and HD. The transcript subgroups of inflammation and epigenetic variables on their own were least successful in classifying disease and healthy control cases in these blood datasets. The nine transcripts selected from transcripts related to stress did make satisfactory classification of ALS and HD from controls. When all 22 transcripts were combined and subjected to LDA to classify disease and controls, we found success with classification within AD, ALS, and HD samples. Our analyses on new publicly available whole blood datasets for our group of previously chosen transcripts demonstrated again the selected transcripts have utility in discriminating disease from healthy controls.

We then used a novel random forest ensemble machine learning algorithm to look for a new set of transcripts in blood that may prove more successful in classifying disease from healthy controls and applied it to all six neurodegenerative diseases. The RF algorithm builds multiple trees to choose the best tree as if assembling a group of weak learners to form a strong learner. We used AD1 and AD2 datasets for training and test sets in both directions and found six transcripts (*MRPL51, NDUFA1, NDUFS5, two Illumina probes for RPL36AL, and LOC646200 (similar to 60S ribosomal protein L22, Heparin binding protein HBp15))* were selected in both sessions, highlighting the importance of mitochondrial and ribosome malfunction in Alzheimer’s disease. The lack of complete agreement for transcript picks by RF for AD1 and AD2 is probably the result of heterogenous disease progression, mixed pathologies, sample and data collection protocols, technical platforms, and processing, highlighting the importance of the standardization of protocols.

Finally, we struggled with the heterogeneities of multiple datasets curated by laboratories around the world on different microarray platforms from two separate manufacturers. We found in this study that a small set of selected transcripts that classify AD from healthy controls also successfully classified ALS and HD from respective controls with excellent accuracy but had poor classification performance in diseases such as PD and FTD, suggesting that there exists sets of transcripts that could classify between multiple neurodegenerative diseases. Machine learning algorithms such as RF are useful but can prove sensitive to microarray technical variations resultant from laboratory collection, compilation, and normalization of data [41]. The RF classification scheme is not limited to two classes. It has been shown to perform accurately in multiclass problems up to 30 classes [42], even when the sample sizes are imbalanced. Datasets of multiple neurodegenerative diseases generated carefully on the same or identical platforms with careful attention to consistency in protocol could reduce technical variations across the disease data or classes. Our ultimate goal would be to work with such datasets of multiple neurodegenerative diseases in which we could identify a small group of transcript estimators to classify between diseases. We opted to work with publicly available data for our analysis as creating multiple sizable datasets would be too costly for this pilot work. A whole blood expression collection method can affect the transcriptomic profile [43] and we selected datasets reporting similar protocols for collection of the whole blood as described in the Methods Section. We downloaded the normalized data deposited by each lab and verified log_2_ normalization. We then applied an additional normalization step to further balance samples using GUSB, a transcript expression that is relatively stable during disease [28,29]. To avoid the complexities and problematic outcomes of merging multiple large datasets, we analyzed each dataset cohort separately before relative comparisons on outcome.

## 5. Conclusions

We show here that blood RNA can be useful to discriminate multiple neurodegenerative diseases from healthy controls. We encountered inconsistencies in analytical data preparation across cohorts, which might explain the small overlap of transcript selection by RF to differentiate disease. For example, two of the three cohorts for AD were collected and analyzed by one research group, but the third AD dataset was assembled by another unrelated laboratory. Regardless, the two sets of selected transcripts for AD were able to differentiate disease from controls across all three cohorts with acceptable statistics. These inconsistencies might not be entirely the result of differences in laboratory instrumentation or protocols but could be explained by substantial preanalytical variations among research cohorts. It is also likely that AD and other diseases have inherent heterogeneity that may even be dynamic during the long preclinical phase of accumulating pathologies before clinical symptoms begin to develop [44].

In summary, we report evidence for developing a noninvasive, broadly accessible blood RNA test platform that can discriminate neurodegenerative disease or provide quick evidence for the enrollment of subjects in clinical trials. Although the scientific community has a long way to go to understand the complexities of cellular processes in humans, blood transcript changes can provide information for dysfunctional pathways in disease and further comparative analysis of blood and brain tissues will be key for the understanding of similarities and differences across multiple neurodegenerative diseases. Successful approaches include the standardization of protocols for collecting and analyzing shared cohort data and the inclusion of experimental hypotheses to explore temporal factors of disease where transcript expression may be different, not only depending on the cell type or tissue region but also because it may be dynamic as disease progresses.

## 6. Patent Applications

Carol Huseby and Paul Coleman filed provisional US patent applications Computer-implemented method and devices for diagnosing neurodegenerative disease (63/143,616) and Algorithm to distinguish multiple neurodegenerative diseases on the basis of transcripts in blood (63/250,889).

## Figures and Tables

**Figure 1 biomolecules-12-01592-f001:**
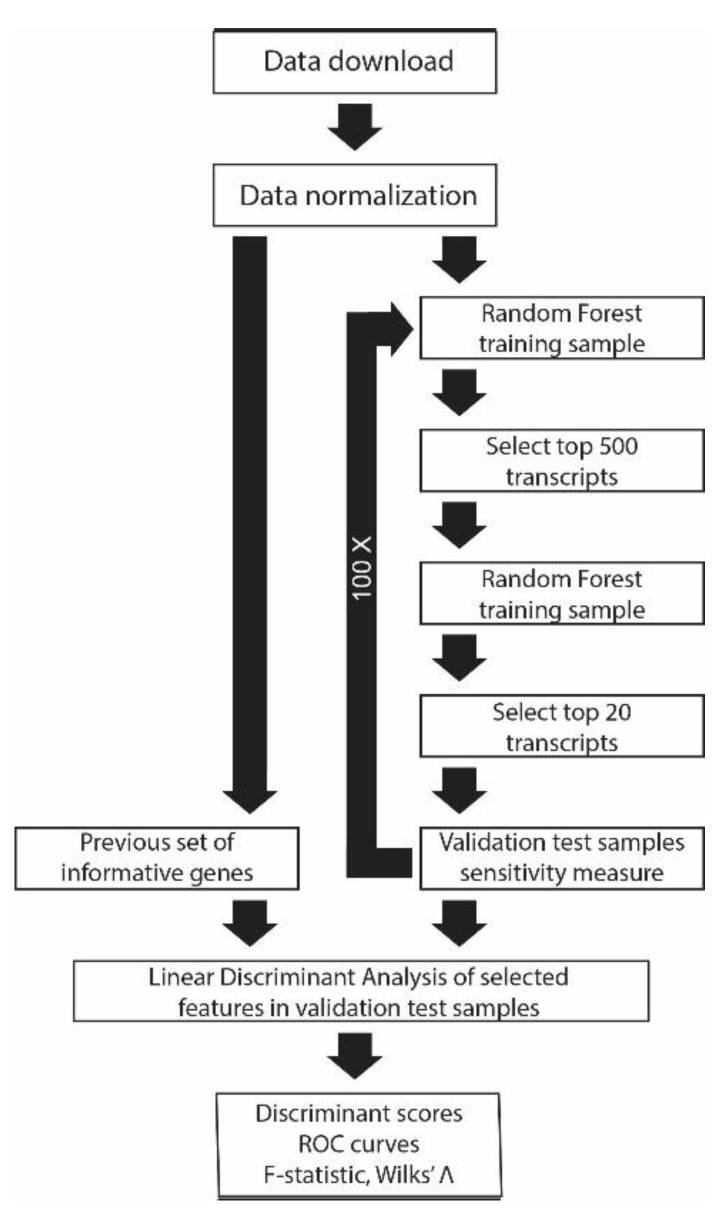
Schematic of analysis method. Selection of predictive transcripts were made ad hoc from empirical data research and subjected to validation by linear discriminant analysis (left side). A machine learning algorithm generated new predictive transcripts for classifying disease from healthy controls (right side), subsequently validated using linear discriminant analysis.

**Figure 2 biomolecules-12-01592-f002:**
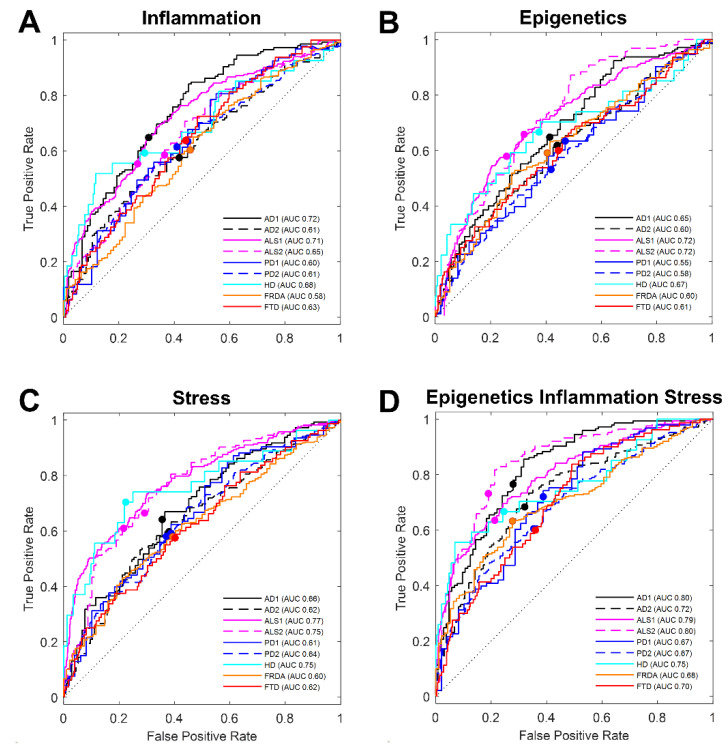
Receiver operator characteristic (ROC) curves for functional group features in AD and other neurodegenerative diseases. ROC curves generated from discriminant scores of linear discriminant analysis using group features selected empirically from literature for each of three functional groups (**A**) inflammation, (**B**) epigenetics, and (**C**) stress. (**D**) ROC curves for combined analysis of all 22 transcripts for the three groups. Line colors denote specific neurodegenerative disease tested and line points mark LDA selected threshold for each. ROC curve line color: AD (black), HD (cyan), FRDA (orange), PD (blue), ALS (pink), and FTD (red). AUC: area under curve probability for predicting classification of disease.

**Figure 3 biomolecules-12-01592-f003:**
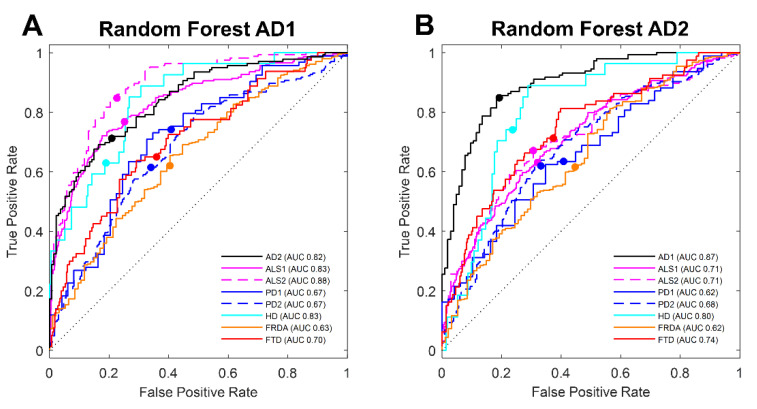
Linear discriminant analysis (LDA) of random forest-selected transcripts from AD whole blood gene expression data. ROC curves generated from discriminant scores of LDA on the top 20 Table 63060. AD1 model #55 with sensitivity 78% (left) and training on (**B**) GSE63061 AD2 model #24 with sensitivity 87% (right). Each line color denotes a specific neurodegenerative disease tested using transcripts selected for AD1 (**A**) or AD2 (**B**). Points on lines mark LDA-selected threshold for each test. ROC curve line color: AD (black), HD (cyan), FRDA (orange), PD (blue), ALS (pink), and FTD (red). AUC: area under curve probability for predicting classification of disease.

**Table 1 biomolecules-12-01592-t001:** List of empirically selected transcripts by molecular functional group.

Inflammation	Epigenetics	Stress
C5	DNMT1	CRYAB
IL10RA	DNMT3A	FTH1
IL17RA	HDAC1	FTL
IL8	HDAC6	GAPDH
LIF	MBD2	HSP90AB1
SERPING1	SIRT1	HSPB1
TNF		PTGS1
		PTGS2
		TFRC

**Table 2 biomolecules-12-01592-t002:** Random Forest disease classifying transcript picks.

GSE63060 AD1	GSE63061 AD2	GSE57475 PD1	GSE99039 PD2	GSE99039 HD	GSE112676 ALS1	GSE112680 ALS2	GSE140830 bvFTD	GSE102008 FRDA
TFDP1	RPS25	ELOVL4	MRPS15	FNDC1	ATP5I	ACAA1	PET100	ABCA1
ATP5I	UFC1	CECR1	CIRBP	SOCS6	ABCA1	ARHGAP30	KLF6	HLA-DRB1
CMTM2	HLA-A/ HLA-A29 allele	PITHD1/ C1orf128	HELZ2/ PRIC285	ANXA2	QPCT	IKBIP/IKIP	SNURF	FKBP1A
DDIT4	HFE/HLA-H	LDLR	MRVI1	FYN	CNPY3	AIF1	UPK3BL	SULT1A1
RPL36AL	CD72	CENPV/ PRR6	SLC35A2	CRK	VIM	SPECC1L/ CYTSA	LCN2	SERPINE2
APBB3	RPL36AL	DHRS4L2	ASXL1	UBE2D3	CTSZ	BRMS1	NGFRAP1	TUBB1
NDUFS5	MS4A7	PPP1R13L	KIR3DL3	DCBLD2	C5AR1	ISG15	DEFA3	NDUFAF3
ING3	UQCRH	MFN2	PTK2B	SESN3	HNRNPUL2	RNF44	POLR1D	NUDT3
GRAP	RPS27A	MEAF6/ C1orf149	FAM102A	HIVEP3	CHKB	HIST1H4C	FYN	RARRES3
SNTB2	DCAF5/ WDR22	SIAH2	SIK3	MCM3	ARF4	PPP3R1	PRDX6	OSCAR
STIP1	NDUFS5	OR51S1	PTGDS	SEPSECS	SLC40A1	MYLPF/ MRLC2	MS4A7	VSTM1
MED16	NDUFA1	TSC22D1	BHLHE40	FECH	CREBBP	ZFP36L2	RUFY1	MRPL2
NDUFA1	C19orf12	HPSE	UBE3A	FANCD2	PPP2R5A	CX3CR1	KLF2	NUDT18
AATF	COTL1	DAPK2	EPB41L2	LONP2	CAPZA2	ATP2B4	ATF6	TAPBP
CDK10	MRPL51	GPR34	LILRB1	DDX42	VPS13C	TMEM131	DUSP1	ZFP36L2
SHFM1	KIAA0907/ KHDC4	GEMIN6	PTGDS	UBFD1	NRIP1	NIN	PPM1F	HPSE
CETN2	RPS25P6	OVCA2	PTGDS	FBXL20	CD82	S100A4	PTCRA	IGFBP3
TPM3	LOC646200	NSUN7	LOC100510080	COL4A3	BRI3	RPS15A	METRNL	VARS2
MRPL51	RPS23P8	THRA	LOC100510377	CENPK	LAMTOR1/ C11orf59	HLA-DRA	FBP1	SASH1
LOC646200	RPL36AL	ENTPD4	IGHG1	CLPTM1	RORA	ZMIZ1	DEFA1B	OSCAR

## Data Availability

Data not contained here including Appendix A are available upon request from C.J.H.

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
