# Peer review of "Blood Transcript Biomarkers Selected by Machine Learning Algorithm Classify Neurodegenerative Diseases including Alzheimer’s Disease"

_biomolecules, 2022, doi:10.3390/biom12111592_

Round 1
Reviewer 1 Report
First of all, I would like to congratulate the authors for this well written manuscript. In myopinion, some issues must be managed to improve the final quality of the paper:
1) Figure 1 is not cle4ar and it seems cuted at the final.
2) MOST IMPORTANT: Why Random Forest? Why don´t SVM or MLP for example?
3) It is necessary write something about the training and the hyperparameters of the ML technique.
Author Response
Please see attachment for Reviewer 1.

Reviewer 2 Report
The manuscript is of great interest: a precise diagnosis of neurodegenerative diseases is very urgent issue todate. The authors present a model which predict different neurodegenerative disorders with high accuracy. The distinctive feature of the manuscript is focusing not only on Alzheimer's disease, but also on other neurodegenerative disorders. However, as different neurodegenerative disorders have overlapping features, the distinguish between different neurodegenerative conditions may be the most difficult part of diagnosis. Thus, my recommendation for authors is to discuss the acceptability of their approaches not only to differentiate between diseaesed people and healthy controls, but also between different types of neurodegenerative diseases.
Besides, there is a minor comment: reference list doesn't match requirements for the formalization; please, see Instructions for authors.
Author Response
Please see attached for Reviewer 2.
